# Drugs in blood and urine samples from victims of suspected exposure to drink spiking: A prospective observational study from Oslo, Norway

Vivian M. Dalaker[1,2], Håvard Furuhaugen[3], Mette Brekke[4], Mari Asphjell Bjørnaas[5], Maja Krpo[6], Elisabeth Leere Øiestad[3,7], Odd Martin Vallersnes[1,2]*

1 Department of General Practice, Institute of Health and Society, University of Oslo, Oslo, Norway, 2 Oslo Accident and Emergency Outpatient Clinic, City of Oslo Health Agency, Oslo, Norway, 3 Section of Drug Abuse Research, Department of Forensic Sciences, Oslo University Hospital, Oslo, Norway, 4 General Practice Research Unit, Institute of Health and Society, University of Oslo, Oslo, Norway, 5 Department of Acute Medicine, Oslo University Hospital Ullevaal, Oslo, Norway, 6 Department of Pharmacology, Oslo University Hospital Ullevaal, Oslo, Norway, 7 Department of Pharmacy, University of Oslo, Oslo, Norway

* o.m.vallersnes@medisin.uio.no

**Data Availability Statement:** Data cannot be shared publicly because public deposition would breach compliance with the protocol approved by

## Abstract

### Objective

People regularly contact emergency medicine services concerned that they have been exposed to drink spiking, i.e., exposure to drugs without their knowledge or permission. We identified drugs in blood and urine samples from patients suspecting exposure to drink spiking, with special consideration for drugs not reported taken by the patient (unreported drugs).

### Methods

From September 2018 to May 2019, we collected blood and urine samples from patients 16 years or older presenting at an emergency clinic in Oslo, Norway, within 48 hours of suspected exposure to drink spiking. We also collected information on ethanol ingestion and drugs taken. Blood samples were analyzed for 20 classical recreational drugs using ultra-high performance liquid chromatography–tandem mass spectrometry (UHPLC-MS/MS) and an automated enzymatic method for ethanol. Urine samples were analyzed using immunoassay methods and a specific gas chromatography mass spectrometry (GCMS) method for gammahydroxybutyrate (GHB).

### Results

From 100 included patients (median age 24 years, 62 females), we collected 100 blood samples and 72 urine samples. Median time since exposure was 5 hours. Unreported drugs were found in 15 patients. Unreported drugs in the blood samples were clonazepam in 3, methylenedioxymethamphetamine (MDMA) in 3, amphetamine in 2, tetrahydrocannabinol (THC) in 2, tramadol in 1, cocaine in 1, and methamphetamine in 1. Unreported drugs in the

our research ethics board, the Regional Committee South-East D for Medical and Health Research Ethics. Inquiries about the data and conditions for access can be made to the corresponding author or to the Institute of Health and Society at the University of Oslo, info@helsam.uio.no.

**Funding:** This work was supported by the Norwegian Research Fund for General Practice. The funder played no role in the research. The funders had no role in study design, data collection and analysis, decision to publish, or preparation of the manuscript.

**Competing interests:** The authors have declared that no competing interests exist.

urine samples were cocaine in 5, amphetamine in 4, ecstasy in 3, and cannabis in 2. Ethanol was found in 69 patients, all reporting ethanol ingestion. Median blood ethanol concentration was higher in patients with no unreported drugs detected, 1.00‰ (interquartile range (IQR) 0–1.52) vs. 0‰ (IQR 0–0.46) (p<0.001). GHB was not detected.

## Conclusion

Unreported drugs, possibly used for drink spiking, were found in 15% of patients. Blood ethanol concentration was higher when no unreported drugs were found. GHB was not detected in any patient.

## Introduction

There has been abundant media coverage in several countries during the recent years focusing on drink spiking [1–4]. People presenting with a concern that someone spiked their drink at a party or in a bar or club is a common occurrence in emergency departments [4–7]. Drink spiking is confirmed in some cases, but sometimes appropriate measures are not taken to investigate the claims. Therefore, the extent of the problem is unknown. Previous studies have found that drink spiking with drugs other than ethanol might not be as widespread as the media coverage suggests [4, 6–9].

Most studies on drink spiking have concentrated on drug facilitated sexual assault [8–11]. In systematic reviews of toxicological analyses from victims of drug facilitated sexual assault, the incidence of detection of different substances varies widely: ethanol was found in 10–66% of patients, cannabis in 1–58%, benzodiazepines in 3–83%, cocaine in 1–37%, amphetamines in 1–20%, methylenedioxymethamphetamine (MDMA) in 1–11%, ketamine in 0–18%, and gammahydroxybutyrate (GHB) in 0–6% [8–11]. Several other drugs were also found. In most studies, context information was sparse, and whether substances were taken voluntarily or not was difficult to discern. In a recent review, 2–22% of cases were estimated to result from covert drug administration [8]. In studies of other drug-facilitated crime, analyses have shown different mixtures of benzodiazepines, and also scopolamine, used to incapacitate and then rob the victims [12, 13].

Some people are concerned about exposure to drink spiking though no other crime has happened. There are few studies on such cases, as these patients are not referred to a sexual assault centre or the police for further forensic tests. An Australian study found nine plausible cases among the 97 alleged drink spiking cases they included, five involving ethanol and one GHB [7]. In a study from inner-city London, ethanol was detected in the biological samples in 90% of the 78 participants, and 60% had a blood alcohol concentration above 1.5 g/L [4]. There were eight possible drink spiking cases, where participants tested positive for drugs they denied intentional exposure to; three involving MDMA, three cannabis, one GHB, and one a benzodiazepine [4]. Among 42 patients presenting to an emergency department in Wales alleging their drink had been spiked, none tested positive for GHB or benzodiazepines, while 65% of those tested had a blood alcohol concentration above 1.60 g/L [5].

Many of the drugs potentially used for drink spiking are rapidly eliminated from the body. GHB is often of concern; a colourless, tasteless liquid that easily can be disguised in someone's drink and is eliminated from the body within 10 hours [14, 15]. Furthermore, varying availability of laboratory testing at emergency departments may lead to delays or to laboratory tests not being taken. Laboratory testing of the suspected spiked drink itself might be a complementary strategy [16]. However, the drink or glass may often not be recoverable.

## Objectives

The aim of this study was to identify drugs in blood and urine samples from patients suspecting exposure to spiked drinks. We considered drugs not reported taken by the patient (unreported drugs) as drugs possibly used to spike a drink.

## Materials and methods

### Design

The study was a prospective observational study. From 1 September 2018 to 31 May 2019, we collected blood and urine samples from patients who presented at the Oslo Accident and Emergency Outpatient Clinic (OAEOC) suspecting exposure to drink spiking.

### Setting

The OAEOC is the main primary care emergency outpatient clinic in Oslo, with approximately 200,000 consultations per year, mainly staffed by general practitioners and nurses. The OAEOC serves the entire city (681,071 inhabitants as per 1 January 2019 [17]) at all hours. Short-time observation and an observation unit with 18 beds are available, with limited diagnostic tools and treatment possibilities. In Oslo, the vast majority of patients with recreational drug toxicity are treated at the OAEOC, but the more severe cases are transferred to hospital or brought directly to hospital by the ambulance service [18].

### Inclusion

Patients 16 years or older themselves raising concern about possible exposure to drink spiking within the last 48 hours were invited to participate in the study. The concern was either spontaneously raised by the patient at presentation or while consulting a doctor after observation for acute poisoning related to recreational drug use. All patients provided written informed consent before inclusion in the study. Patients who were unconscious or too intoxicated to provide informed consent were invited after having regained their capacity to consent after treatment and/or observation. Treatment was provided according to local procedure [18]. Victims of sexual assault were immediately transferred to the Sexual Assault Centre and not included in the study, as blood and urine sampling for these patients require procedures for maintaining a forensic chain of evidence.

### Data collection

Blood and urine samples were obtained by a nurse, and information was collected through a questionnaire administered by the doctor treating the patient.

The patients provided information of intake of alcohol, recreational drugs, and prescription drugs (including prescribed opioids, benzodiazepines, and methylphenidate) during the last 48 hours (asked as an open question, no predefined categories), and details of the symptoms and clinical features that led to concern about exposure to drink spiking (amnesia, different drug effect experience than expected, more/less hangover, or other causes). Details concerning the time, place, and observations by relatives or friends were recorded. We also registered age, gender, time (or time period) for ingestion of a possibly spiked drink, symptoms at presentation, clinical features at presentation (level of consciousness (Glasgow Coma Scale score), heart rate, blood pressure, temperature, and respiratory rate), and treatment. We also registered the results of any supplementary investigations, e.g., electrocardiogram or other blood samples.

## Toxicological sampling and analyses

Venous blood samples were collected in serum tubes (BD Vacutainer) and stored in a refrigerator at 4˚C before being transported in a cooler bag approximately once a week to the Section of Drug Abuse Research, Department of Forensic Sciences, Oslo University Hospital, for analysis. In a few cases where transport was not possible within one week, the blood samples were temporarily frozen. Whole blood concentrations were determined using ultra-high performance liquid chromatography–tandem mass spectrometry (UHPLC-MS/MS) for amphetamine, methamphetamine, MDMA, cocaine, benzoylecgonine, tetrahydrocannabinol (THC), alprazolam, diazepam, flunitrazepam, clonazepam, nitrazepam, oxazepam, zolpidem, zopiclone, buprenorphine, codeine, methadone, morphine, and tramadol [19]. In brief, samples were prepared using supported liquid extraction with ethylacetate:heptane (80:20 v:v) and analysed using a Waters Aquity UPLC® HSST3 column, 2.1 x 100 mm, 1.8 µm particles (Waters, MA, USA) using methanol and 10 mM ammonium formate buffer pH 3.1 as mobile phase. Mass detection was performed in positive electrospray mode on a Waters TQS instrument (Waters, MA, USA). Ethanol (alcohol) was analysed on an AU680 clinical chemistry analyzer (Beckman Coulter, Brea, CA, USA) by an automated enzymatic method using alcohol dehydrogenase [20]. A modification of a previously published method [21] was used for GHB in blood, only the sample preparation was done using Captiva Lipid ND 96-well plates in stead of OASIS HLB 96-well extraction plates. A similar instrumental set up to the general drug determination using a T3 column and a Xevo TQS instrument was applied, however 0.2% formic acid and methanol were used as mobile phase constituents. Reporting limits in the blood samples are shown in S1 Table in S1 File.

The urine samples (collected in Urin-Monovette with no additives) were stored in a refrigerator at 4˚C before being sent on to the The Department of Pharmacology, Oslo University Hospital Ullevaal for screening analysis every 24 hours. An immunoassay screening method based on spectrophotometry was used and performed on the AU680 clinical chemistry analyzer (Beckman Coulter, Brea, CA, USA). Urine samples were screened for amphetamines (EMIT d.a.u. reagents from Siemens, SYVA), benzodiazepines, opiates, cocaine, THC and metabolites (CEDIA TDM reagents from Thermo Fisher Scientific), ethanol, and ecstasy (MDMA, methylenedioxyamphetamine (MDA), methylenedioxyethylamphetamine (MDEA), methylbenzodioxolylbutanamine (MBDB), benzodioxolylbutanamine (BDB), paramethoxyamphetamine (PMA), and paramethoxymethamphetamine (PMMA)) (DRI TDM reagents from Thermo Fisher Scientific, Waltham, MA, USA). Measurements of pH and creatinine were performed. GHB was tested for in all urine samples, using a specific gas chromatography mass spectrometry (GC-MS) method. GHB analysis was performed by headspace GC coupled to a mass spectrometer. The system consisted of a 7697 Headspace Sampler, a 7890B GC system and a 5977A MSD mass spectrometer, all from Agilent Technologies (Santa Clara, CA, USA). GHB was converted to gammabutyrolactone (GBL) by using sulphuric acid as a catalyst in a headspace vial. The MS is operated in EI mode (70 eV), and GBL was determined using m/z 86 (quantifier ion) and m/z 56 (qualifier ion). α-methyl GBL (AMGBL) was used as an internal standard. The headspace and GC settings were: transfer line at 280˚C, loop at 250˚C, oven temperature 170˚C isothermal, carrier gas helium at flow 1 ml/min, column DB-624 UI. Cut-off values for the urine samples are shown in S2 Table in S1 File.

Information about the toxicological results was provided to patients by the study investigator (VMD) by telephone. At the same time the previously collected information was confirmed or corrected.

## Outcome measures

Unreported drugs were defined as drugs found in blood and/or urine not reported taken by the patient within the previous 48 hours. We considered the unreported drugs as drugs possibly used to spike a drink.

## Ethics

The study was performed in accordance with the Helsinki declaration. Participants provided written informed consent. The study was approved by the Oslo University Hospital Information Security and Privacy Office and by the Regional Committee South-East for Medical and Health Research Ethics (REK sør-øst D 2017/1880).

## Study registration

The study was registered at ClinicalTrials.gov (NCT03651778).

## Statistics

Analyses were done in IBM SPSS versions 28–29 (IBM Corp, Chicago, IL). Pearson's chi-square test or Fisher's exact test (for expected cell counts of five or less) were used to compare frequencies. Mann–Whitney U-test was used in comparisons of continuous variables. Missing data were not included in the analyses.

# Results

During 37 weeks, 100 patients (2.7 per week) were included. Median age was 24 years (range 16–63), 62 were females, 38 were males. We collected 100 blood samples and 72 urine samples. Unreported drugs were found in 15 patients.

Reasons reported for suspecting exposure to drink spiking were the same whether unreported drugs were found or not (Table 1). Median time from ingestion of a suspected spiked drink to biological sampling was 12 hours when unreported drugs were found, and 4 hours when not (p = 0.02). We obtained blood samples from 48 of the patients within 4 hours (urine from 35), from 74 patients within 12 hours (urine from 56), and from 95 patients within 24 hours (urine from 70). The most commonly reported symptoms were amnesia (63%), vomiting (20%), nausea (14%), feeling intoxicated/drunk (13%), and dizziness (12%).

Among the 91 patients reporting having ingested ethanol, it was detected in blood in 66 (among them also in urine in 46), and in urine only in another three. Median blood ethanol concentration was higher when no unreported drugs were found, 1.00‰ (interquartile range (IQR) 0–1.52) vs. 0‰ (IQR 0–0.46) (p<0.001) (Table 1). In general, blood ethanol concentration was lower the longer time since the incident (Fig 1). Median blood ethanol concentration, when detected, was 1.25‰ (IQR 0.90–1.61; range 0.03–3.47). Ethanol was not detected in either matrix in any of the nine patients not reporting ingestion, amongst whom six presented within three hours of the incident.

The unreported drugs found in the blood samples were clonazepam in 3 (3%), MDMA in 3 (3%), amphetamine in 2 (2%), THC in 2 (2%), tramadol in 1 (1%), cocaine in 1 (1%), and methamphetamine in 1 (1%) (Tables 2 & 3). The unreported drugs found in the urine samples were cocaine in 5 (7%), amphetamine in 4 (6%), ecstasy in 3 (4%), and cannabis in 2 (3%). Not counting ethanol, more than one drug was found in 7 cases, all of them among the cases with unreported drugs. There were some discrepancies between findings in blood and urine (Table 3). GHB was not detected in either blood or urine samples.

**Table 1. Reasons for suspecting exposure to drink spiking, symptoms at presentation, and clinical observations.**

| | Cases with any drug not reported taken by the patient | Cases with only drugs reported taken by the patient | Total |
|---|---|---|---|
| | n (%) / median (IQR) | n (%) / median (IQR) | n / median (IQR) |
| **Sex** | | | |
| *Females* | 8 (53) | 54 (64) | 62 |
| *Males* | 7 (47) | 31 (37) | 38 |
| **Age** (years) | 27 (22–32) | 23 (20–30) | 24 (20–31) |
| **Reasons for suspecting exposure to drink spiking** | | | |
| *Different drug effect experience* | 11 (73) | 57 (67) | 68 |
| *Amnesia* | 7 (47) | 56 (66) | 63 |
| *More/less hangover* | 6 (40) | 18 (21) | 24 |
| *Other reason* | 9 (60) | 42 (49) | 51 |
| **Symptoms reported** | | | |
| *Vomiting* | 3 (20) | 17 (20) | 20 |
| *Nausea* | 3 (20) | 11 (13) | 14 |
| *Feeling drunk/intoxicated* | 3 (20) | 10 (12) | 13 |
| *Dizziness* | 4 (27) | 8 (9) | 12 |
| **Clinical observations at presentation** | | | |
| *Glasgow Coma Scale score* | 15 (15–15)[a] | 15 (15–15)[b] | 15 (15–15)[b] |
| *Respiratory rate (per minute)* | 15 (14–17) | 16 (14–17) | 16 (14–17) |
| *Heartrate (beats per minute)* | 96 (77–106) | 86 (76–98) | 87 (76–97) |
| *Systolic blood pressure (mmHg)* | 131 (118–137) | 121 (114–131) | 122 (114–131) |
| *Temperature (°C)* | 36.5 (36.2–36.9) | 36.3 (35.7–36.7) | 36.3 (35.7–36.7) |
| **Time since suspected exposure** (hours) | 12 (4–24) | 4 (2–12) | 5 (2–14) |
| **Serum ethanol concentration** (‰) | 0 (0–0.46) | 1.00 (0–1.52) | 0.89 (0–1.42) |
| **Number of drugs other than ethanol** | | | |
| *1* | 8 (53) | 5 (6) | 13 |
| *2* | 5 (33) | - | 5 |
| *≥3* | 2 (13) | - | 2 |
| **Total** | 15 (100) | 85 (100) | 100 |

IQR: interquartile range.

No statistically significant differences between groups, except for time since suspected exposure p = 0.033, serum ethanol p<0.001, and number of drugs found p<0.001.

[a]All patients had Glasgow Coma Scale score 15.

[b]Glasgow Coma Scale score range 12–15.

Sixty-four (64%) patients were discharged without any observation or treatment, 30 (30%) were kept for observation for some hours (median 3 hours, range 1–4), 10 (10%) were given symptomatic treatment, none were given antidote, and 6 (6%) were transferred to hospital due to need for further treatment of poisoning. No patients died.

## Discussion

Unreported drugs were found in 15 of the 100 patients and included clonazepam, tramadol, MDMA, cocaine, amphetamine, methamphetamine, and THC. GHB was not detected. Ethanol was detected in blood in 66 of the 91 patients reporting ingestion, and in urine only in another three. Blood ethanol concentration was higher when no unreported drugs were found.

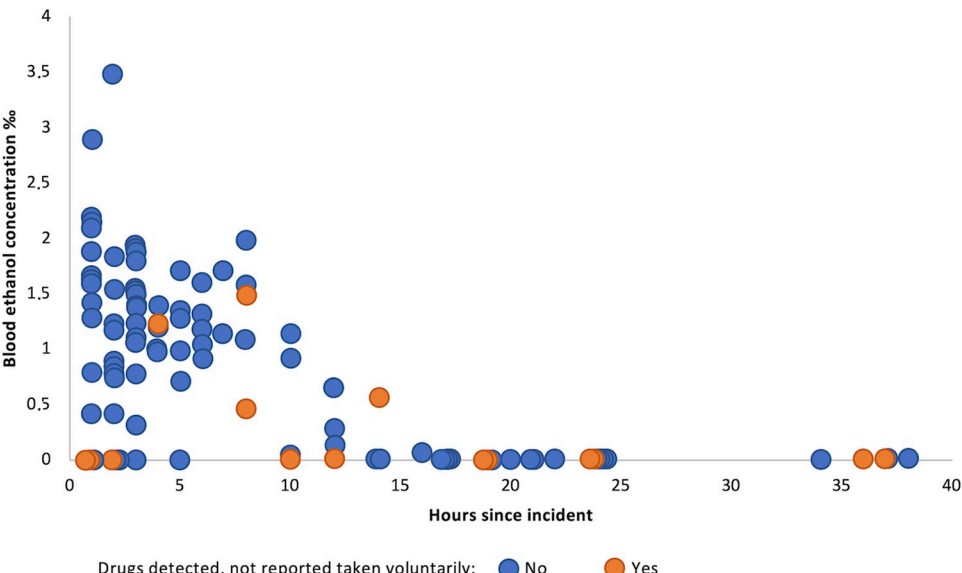

**Fig 1. Blood ethanol concentration and time since suspected exposure to drink spiking.**

## Possible drink spiking cases

We found unreported drugs in only 15% of the patients suspecting exposure to drink spiking. This is in accordance with the estimate of 2–22% in a recent review [8]. However, cases of drink spiking may have gone undetected, as drugs we did not screen for may be used to spike drinks. We did not analyze for antihistamines, antidepressants, or ketamine, drugs not uncommonly detected in victims of drug facilitated sexual assault [8, 10]. Nor did we analyze for any of the numerous novel psychoactive substances that have appeared during the last two decades, also having a potential to be used for drink spiking [22–25]. Furthermore, a drug taken by the patient may also have been used for drink spiking, adding to the effect of the drug voluntarily taken. Hence, our study probably underestimates the proportion actually subjected to drink spiking among those raising concern.

Clonazepam was detected in three cases, the only benzodiazepine found among the unreported drugs in our study. This is surprisingly few considering how frequently benzodiazepines are detected in drug-facilitated sexual assault [8], though in line with the few reports from other emergency department settings [4, 5, 7]. Benzodiazepines are CNS depressants controlled under drug or medicine legislation and can induce confusion, impaired thinking, memory loss, drowsiness, sleepiness, fatigue, impaired coordination, and dizziness [26]. Flunitrazepam (marketed under the name Rohypnol) has previously been associated with drink spiking, prompting the manufacturer (Roche Pharmaceuticals, Basel, Switzerland) to modify the product formulation and add a colourant, a blue dye fizzing in liquids [10]. Flunitrazepam has been deregistered in Norway but is still used illegally [27]. Clonazepam and diazepam currently dominate in recreational drug toxicity and driving under the influence of drugs cases in Norway [27, 28].

Surprisingly, GHB was not detected in any of the samples, although it is often suspected in drink spiking cases, both by patients and by doctors [5, 10]. The window of detection is relatively short; GHB is eliminated from plasma with a half-life of 30–50 minutes and is detectable in urine for 3 to 10 hours [15]. More than half of our patients had samples taken within the window of detection, as we obtained blood samples from 48 of the patients within 4 hours, and

**Table 2. Substances found in blood and urine samples in patients suspecting exposure to drink spiking.**

| | Samples with any substance not reported taken by the patient | Samples without substances not reported taken by the patient | Total samples with substance |
|---|---|---|---|
| | n (%) | n (%) | n (%) |
| **Blood** | | | |
| *Any substance* | 12 (100)[a] | 69 (78)[a] | 75 (75)[a] |
| *Clonazepam* | 3 (25) | - | 3 (3) |
| *MDMA* | 3 (25) | - | 3 (3) |
| *Amphetamine* | 2 (17) | 1 (1) | 3 (3) |
| *THC* | 2 (17) | 1 (1) | 3 (3) |
| *Cocaine* | 1 (8) | 1 (1) | 2 (2) |
| *Tramadol* | 1 (8) | - | 1 (1) |
| *Methamphetamine* | 1 (8) | - | 1 (1) |
| *Ethanol* | - | 66 (75) | 66 (66) |
| *Alprazolam* | - | 1 (1) | 1 (1) |
| **Total blood samples** | 12 (100) | 88 (100) | 100 (100) |
| **Urine** | | | |
| *Any substance* | 10 (100)[a] | 53 (85)[aa] | 56 (78)[a] |
| *Cocaine* | 5 (50) | 2 (3) | 7 (10) |
| *Amphetamine* | 4 (40) | 1 (2) | 5 (7) |
| *Ecstasy*[b] | 3 (30) | - | 3 (4) |
| *Cannabis* | 2 (20) | 4 (6)[c] | 6 (8) |
| *Ethanol* | - | 49 (79) | 49 (68) |
| *Benzodiazepines* | - | 1 (2) | 1 (1) |
| *Opiates* | - | 1 (2) | 1 (1) |
| **Total urine samples** | 10 (100) | 62 (100) | 72 (100) |

[a]Any substance total is less than the sum of individual substances, as several drugs were detected in several cases, cf. Table 3.

[b]MDMA and related drugs.

[c]In all 4 cases, cannabis not reported used last 48 hours; found in urine, not in blood. Probably metabolites from previous cannabis use.

MDMA: methylenedioxymethamphetamine; THC: tetrahydrocannabinol.

urine samples from 56 patients within 12 hours. In a similar study in London, GHB was detected in blood and urine in only one of 78 participants, who had ingested what he thought was an alcoholic drink at a club [4]. Likewise, GHB used for drink spiking was only found in one of 101 patients in a similar Australian study [7].

MDMA was detected in four drink spiking cases, mostly combined with other drugs and/or ethanol. In recent years the popularity of this drug has increased in Europe [29]. MDMA is an empathogen with some stimulant effects, in itself not associated with sedation, but might lead to disinhibition, and for that reason it is conceivable that this drug can be used to spike someone's drink [4, 30]. We also found cocaine, amphetamine, and methamphetamine in some drink spiking cases, often combined with each other, alcohol and/or MDMA, similar to findings from the UK [4, 5]. Stimulants do not lead to memory loss, but as other recreational drug they have disinhibiting effects that lead to impaired control, which is a risk factor for drug facilitated sexual assault [30].

Unreported cannabis was detected in urine in six cases, in two of them also in blood, all samples obtained within 19 hours of exposure. In two cases, cannabis only found in urine was the only unreported drug. THC is usually not detectable in blood 24 hours after intake, but

**Table 3. Patients suspecting exposure to drink spiking, with drugs other than ethanol detected in blood and/or urine sample (n = 19).**

| Gender/ age | Substance | Blood concentration nM | Detected in urine | Time since suspected exposure (hours) | Drug not reported taken | Symptoms and clinical features |
|---|---|---|---|---|---|---|
| M26[a] | Clonazepam | 282 | | 2 | X | Dizzy, tired, fainting, amnesia, different drug effect experience than expected |
| | Amphetamine | 3131 | Yes | | X | |
| | Alprazolam | 139 | | | | |
| | Benzodiazepines | | Yes | | | |
| | Opiates | | Yes | | | |
| | Cannabis | | Yes | | | |
| F21[a] | Clonazepam | 20 | | 1 | X | Paranoid, apparently intoxicated |
| | Amphetamine | 1692 | Yes | | | |
| | Cannabis | | Yes | | | |
| M17[b] | Clonazepam | 99 | | 1 | X | Body pain |
| F29[b] | Tramadol | 211 | No urine sample | 24 | X | Different drug effect experience than expected, more hangover, felt unwell, tired, stomach pain, diarrhoea, nausea, vomiting, fever, disoriented, fainted |
| F21 | MDMA | 4312 | Yes | 12 | X | Different drug effect experience than expected, unwell, nauseous |
| | Amphetamine | | Yes | | X | |
| F29 | MDMA | 968 | Yes | 19 | X | Drank water, immediately unwell, vomiting, disoriented, paranoid, mydriasis, then miosis, iris colour changed |
| | Amphetamine | | Yes | | X | |
| F51[b] | MDMA | 211 | No urine sample | 24 | X | Different drug effect experience than expected, more hangover, dizzy, cold, difficult to move |
| M32[b] | Methamphetamine | 790 | No urine sample | 19 | X | Amnesia, different drug effect experience than expected, paranoid, stressed, overactivated, restless, afraid to die, suicidal thoughts |
| | Cannabis | 3 | | | | |
| F28 | Amphetamine | 436 | Yes | 8 | X | Hectic, agitated, thirsty, stressed, restless, mydriasis, thirsty, jaw pain |
| | Cocaine | | Yes | | X | |
| | Ethanol | 1.47 | Yes | | | |
| M37 | Cocaine | 53 | Yes | 10 | X | Amnesia, different drug effect experience than expected, felt intoxicated, visual disturbances |
| | Benzoylecgonine | 1529 | | | X | |
| | Ethanol | | Yes | | | |
| M25 | Cocaine | | Yes | 8 | X | Amnesia, different drug effect experience than expected, more hangover, felt unwell, unfocused |
| | Ethanol | 0.46 | Yes | | | |
| F22[b] | Cocaine | | Yes | 37 | X | Amnesia, more hangover, vomiting, chest pain, breathlessness |
| M27[b] | Cocaine | | Yes | 36 | X | Amnesia, different drug effect experience than expected, felt warm, blackout, strange behaviour, tired, felt distant and unwell |
| | MDMA | | Yes | | X | |
| F35 | Cannabis | 3 | Yes | 14 | X | Amnesia, different drug effect experience than expected, more hangover, palpitations, felt dizzy and strange |
| | Ethanol | 0.55 | Yes | | | |
| M24 | Cannabis | 3 | Yes | 4 | X | Trembling, felt unwell, nauseous |
| | Ethanol | 1.24 | Yes | | | |
| M29 | Cocaine | 94 | Yes | 1 | | Arms felt paralyzed |
| | Benzoylecgonine | 238 | | | | |
| | Cocaethylene | 63 | | | | |
| | Ethanol | 1.63 | Yes | | | |
| F30[b] | Cocaine | | Yes | 14 | | Different drug experience than expected, felt powerless, not able to move, headache, tired |
| M33[a] | Cannabis | | Yes | 10 | | Different drug effect experience than expected, more hangover, felt distant, diplopia, behaved differently |
| | Ethanol | 1.14 | Yes | | | |

*(Continued)*

**Table 3.** (Continued)

| Gender/ age | Substance | Blood concentration nM | Detected in urine | Time since suspected exposure (hours) | Drug not reported taken | Symptoms and clinical features |
|---|---|---|---|---|---|---|
| F20[a,b] | Cannabis | | Yes | 19 | | Amnesia |

[a]Cannabis not reported used last 48 hours; found in urine, not in blood. Probably metabolites from previous cannabis use.

[b]Ethanol reported ingested, but not found in blood or urine.

Cannabis: tetrahydrocannabinol in blood; tetrahydrocannabinol and metabolites in urine.

MDMA: methylenedioxymethamphetamine in blood; methylenedioxymethamphetamine and related substances in urine (methylenedioxyamphetamine (MDA), methylenedioxyethylamphetamine (MDEA), methylbenzodioxolylbutanamine (MBDB), benzodioxolylbutanamine (BDB), paramethoxyamphetamine (PMA), and paramethoxymethamphetamine (PMMA)).

metabolites can be detected in urine for weeks after use [31]. It is highly likely that the cannabis only detected in urine samples stemmed from cannabis use more than 48 hours before presentation and hence not reported, rather than covert cannabis administration. Accordingly, we did not consider cannabis only detected in urine an unreported drug.

## Ethanol

Nearly all patients reported ingesting ethanol. One out of four patients presented 14 hours or later after the suspected time of exposure to drink spiking, which is probably why ethanol in many cases was not detected (Fig 1). Ethanol has a sedative effect that causes loss of self-control and impaired consciousness, and the effect is enhanced when combined with drugs. The possibility that ethanol, rather than another drug, is used to spike an alcoholic drink to amplify the sedative effect, is also a plausible explanation for intoxication that does not correspond to what people thought they had been drinking. Our finding that blood ethanol concentrations were higher when no unreported drugs were detected might result from drink spiking with ethanol. On the other hand, these patients presented earlier, which also may explain the higher blood ethanol concentrations. It is also possible that people may underestimate the amount or the effect of the alcohol they have ingested. In a study among 264 patients at a Norwegian sexual assault centre, covert drug administration was more often suspected by the patient the higher the blood ethanol concentration [32]. In our study one out of four had 1.42‰ or above, in line with other studies, emphasizing high ethanol concentrations as a possible explanation for the patient's condition [4, 5, 8, 32]. Whether the ethanol was voluntarily ingested or covertly administered remains hard to prove.

## Testing for drink spiking

Suspected exposure to drink spiking is a major concern for the victim, also in the absence of any additional crime. Most people would like to know whether an exposure actually happened. However, there are significant barriers to having this service available, including timing, cost, and providing analysis that covers for the many hundred possible drugs. Limited analytical libraries may give false negative results. Keeping up with the expanding abundance and variety of the novel psychoactive substances is a challenge to clinicians, developers of toxicological tests, and legislatures [25, 33, 34]. More importantly, it constitutes an additional health hazard as users are often not aware of the potential dangers of the novel drug at hand [25]. The situation calls for comprehensive international efforts in toxicovigilance, information exchange, and regulative measures [25, 33].

Another issue is the appropriateness of using emergency departments for this service as most patients were clinically well when they presented. Furthermore, in many jurisdictions, emergency department analyses may not be admissible in a court of law. Still, the service is clearly important to the public [1–3].

## Strengths and limitations

Our study is one of very few exploring suspected drink spiking cases in the absence of any additional crime.

There may be some uncertainty in the information regarding the reported intake of ethanol and/or drugs and time of suspected exposure to drink spiking, due to intoxication, amnesia or confusion, but most of the patients have confirmed or corrected their information through a telephone consultation conducted when the test results were available. At this time, patients could also confirm or correct which drugs they had taken, both recreational drugs and prescription drugs. However, we must take into account that some might not want to inform about the use of drugs or did not know that drugs taken days or weeks ago still could be detected, such as cannabis in urine.

As far as we know, all patients raising concern about exposure to drink spiking were offered participation in the study. At the time, laboratory toxicological testing was not part of standard care for patients not reporting sexual assault. Consequently, it is highly likely that nearly all eligible patients were included. After the study period, toxicological testing in urine samples was made standard care.

Some cases may have been missed because the patient was brought directly to the hospital by ambulance for treatment due to severe toxicity, bypassing the OAEOC. Pointing in the same direction, most of our patients had a Glasgow Coma Scale score of 15 at presentation. Hence, our population of relatively not severely sick patients may not be representative for the larger population of patients with suspected drink spiking.

The analysis of both blood and urines samples were done using validated methods. However, we did not follow procedures for maintaining a forensic chain of evidence. This may partially explain some of the missing urine samples. We obtained blood samples from all of the 100 patients included, but urine samples from only 72. Some of the patients were not able to provide a urine sample, and some samples were handled inappropriately, and in a few cases lost during transport.

The blood samples were transported once a week to the laboratory. Hence, more than three days might elapse between sampling and analysis, yielding reduced GHB levels [35]. However, it is unlikely that the storage time should reduce the GHB levels below the level of detection. The lack of detection of GHB in the urine samples, transported to the laboratory and analyzed at a daily basis, further supports the validity of this result.

## Conclusions

We found a variety of drugs reported taken by the patient. Unreported drugs were found in one of six cases, probably representing drink spiking with benzodiazepines, tramadol, MDMA, amphetamines, cocaine, and cannabis. We found no GHB. Ethanol was widely reported, and blood alcohol concentrations were high among some. Whether ethanol was used to spike alcoholic drinks in our patients is impossible to assess, but clinicians should be observant to the possibility.

Drink spiking is of serious concern to individuals thinking themselves possibly exposed. Laboratory testing can help to some extent, though limited analytical libraries and timing issues are a problem. Furthermore, drinks may be spiked with ethanol, or with a drug also

taken voluntarily by the patient. Then, there is the possibility that the patient underestimates the effects of ethanol and/or drugs taken intentionally. All these factors should be considered when addressing the concern of the patient suspecting exposure to drink spiking.

## Supporting information

**S1 Checklist. STROBE statement—Checklist of items that should be included in reports of *cross-sectional studies*.**
(DOC)

**S1 File. S1 and S2 Tables.** Reporting limits in blood samples and cut-off values in urine samples.
(PDF)

## Acknowledgments

Our colleagues at the OAEOC are gratefully acknowledged for including the patients in the study and collecting the blood and urine samples.

This work was performed on the Services for sensitive data (TSD) facilities, owned by the University of Oslo, operated and developed by the TSD service group at the University of Oslo, IT-Department (USIT), (tsd-drift@usit.uio.no).

## Author Contributions

**Conceptualization:** Vivian M. Dalaker, Odd Martin Vallersnes.

**Data curation:** Vivian M. Dalaker, Odd Martin Vallersnes.

**Formal analysis:** Vivian M. Dalaker, Odd Martin Vallersnes.

**Funding acquisition:** Vivian M. Dalaker.

**Investigation:** Vivian M. Dalaker, Håvard Furuhaugen.

**Methodology:** Vivian M. Dalaker, Håvard Furuhaugen, Mette Brekke, Mari Asphjell Bjørnaas, Odd Martin Vallersnes.

**Project administration:** Vivian M. Dalaker.

**Resources:** Håvard Furuhaugen, Maja Krpo, Elisabeth Leere Øiestad.

**Supervision:** Odd Martin Vallersnes.

**Validation:** Håvard Furuhaugen, Maja Krpo, Elisabeth Leere Øiestad, Odd Martin Vallersnes.

**Visualization:** Vivian M. Dalaker, Odd Martin Vallersnes.

**Writing – original draft:** Vivian M. Dalaker, Odd Martin Vallersnes.

**Writing – review & editing:** Vivian M. Dalaker, Håvard Furuhaugen, Mette Brekke, Mari Asphjell Bjørnaas, Maja Krpo, Elisabeth Leere Øiestad, Odd Martin Vallersnes.

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
