## [Decision Letter · Decision Letter 0]

9 Jan 2024

PONE-D-23-26541Drugs in blood and urine samples from suspected spiked drink victims: a prospective observational study from Oslo, NorwayPLOS ONE

Dear Dr. Vallersnes,

Thank you for submitting your manuscript to PLOS ONE. After careful consideration, we feel that it has merit but does not fully meet PLOS ONE’s publication criteria as it currently stands. Therefore, we invite you to submit a revised version of the manuscript that addresses the points raised during the review process.

I would like to sincerely apologise for the delay you have incurred with your submission. It has been exceptionally difficult to secure reviewers to evaluate your study. We have now received four completed reviews; the comments are available below. Please pay particular attention to Reviewer#4 comments, this reviewer has raised significant scientific concerns about the study that need to be addressed in a revision.

Please revise the manuscript to address all the reviewer's comments in a point-by-point response in order to ensure it is meeting the journal's publication criteria. Please note that the revised manuscript will need to undergo further review, we thus cannot at this point anticipate the outcome of the evaluation process.

We look forward to receiving your revised manuscript.

Kind regards,

Miquel Vall-llosera Camps

Staff Editor

PLOS ONE

Comments from PLOS Editorial Office:

We note that one or more reviewers has recommended that you cite specific previously published works. As always, we recommend that you please review and evaluate the requested works to determine whether they are relevant and should be cited. It is not a requirement to cite these works. We appreciate your attention to this request.

Reviewers' comments:

Reviewer's Responses to Questions

**Comments to the Author**

1. Is the manuscript technically sound, and do the data support the conclusions?

Reviewer #1: Yes

Reviewer #2: Yes

Reviewer #3: Yes

Reviewer #4: Partly

2. Has the statistical analysis been performed appropriately and rigorously? 

Reviewer #1: Yes

Reviewer #2: Yes

Reviewer #3: Yes

Reviewer #4: N/A

3. Have the authors made all data underlying the findings in their manuscript fully available?

Reviewer #1: Yes

Reviewer #2: Yes

Reviewer #3: Yes

Reviewer #4: No

4. Is the manuscript presented in an intelligible fashion and written in standard English?

Reviewer #1: Yes

Reviewer #2: Yes

Reviewer #3: Yes

Reviewer #4: No

5. Review Comments to the Author

Reviewer #1: Dear Authors,

It was my pleasure to review the article titled Drugs in blood and urine samples from suspected spiked drink victims: a prospective

observational study from Oslo, Norway, a solid and well researched article centered around drug-facilitated sexual assault (or less commonly other crimes targeting an uncconscious ibìncapacitated victim). The study encompasses a 100-patient sample and focuses on detection of unreported drugs.

The article has noteworthy strengths: it is thorough in terms of pursuing its stated objective; it is relevant and overall a worthy contribution to toxicology research; it has an element of novelty in its design, and relies on sound methodology as far as I was able to determine.

It is my belief that the article, particularly in its discussion, could benefit from a slightly higher degree of contextualization and broader scope when it comes to screening and detection techniques. It would certainly make the article more comprehensive and well-rounded to draw comparisons and outline distinctive features of various techniques and their analytical value in terms of tackling the spread of illegal/misused substances, both for recreational use and as a tool to victimize others. Broader remarks on the legal and law enforcement potential of detection efforts, at least within the European context, could also be advisable. The following sources should be drawn upon and cited:

Brunetti P, Lo Faro AF, Di Trana A, Montana A, Basile G, Carlier J, Busardò FP. β'-Phenylfentanyl Metabolism in Primary Human Hepatocyte Incubations: Identification of Potential Biomarkers of Exposure in Clinical and Forensic Toxicology. J Anal Toxicol. 2023 Jan 24;46(9):e207-e217. doi: 10.1093/jat/bkac065.

Di Trana A, Brunetti P, Giorgetti R, Marinelli E, Zaami S, Busardò FP, Carlier J. In silico prediction, LC-HRMS/MS analysis, and targeted/untargeted data-mining workflow for the profiling of phenylfentanyl in vitro metabolites. Talanta. 2021 Dec 1;235:122740. doi: 10.1016/j.talanta.2021.122740.

Lo Faro AF, Venanzi B, Pilli G, Ripani U, Basile G, Pichini S, Busardò FP. Ultra-high-performance liquid chromatography-tandem mass spectrometry assay for quantifying THC, CBD and their metabolites in hair. Application to patients treated with medical cannabis. J Pharm Biomed Anal. 2022 Aug 5;217:114841. doi: 10.1016/j.jpba.2022.114841.

Busardò FP, Zaami S, Baglio G, Indorato F, Montana A, Giarratana N, Kyriakou C, Marinelli E, Romano G. Assessment of the stability of exogenous gamma hydroxybutyric acid (GHB) in stored blood and urine specimens. Eur Rev Med Pharmacol Sci. 2015 Nov;19(21):4187-94.

The tables and figures are qite well crafted and effective at conveying key points and relevant findings.

Although the article is well-written overall, I recommend further proofreading by a native speaker of English.

Reviewer #2: The article is competently assembled and straightforward enough overall. Its aim is clearly delineated and the methodology appears to be sound. tables and figures contribute to the conveyance of key elements providing substance and clarity to the article' s fundamental reasoning and conclusions.

The Discussion I believe should be further developed to better highlight the article's relevance in toxicology research. More meaningful sources ought to be used in order to better elaborate on detection techniques and their value in tackling substance abuse and in upholding public health. The following should be looked at and cited in that regard: DOI: 10.23750/abm.v92i6.12696; DOI 10.1016/j.jpba.2020.113335. Furthermore, I suggest the authors brefly address legislative aspects as well against the backdrop of the major threat posed by novel psychoactive substances DOI: 10.1002/hup.2727; DOI: 10.1007/164_2018_160; DOI: 10.26355/eurrev_201911_19529). Such additions will provide an extra degree of elaboration which will highlight the importance and value of the article's conclusions and findings.

Sincerely.

Reviewer #3: Comments:

1. In abstract (line no 47), Check the sentence consistency.

2. Remove drink spiking word from the keyword.

3. Literature survey regarding analysis of drugs in case of drug facilitated crimes (DFC) is insufficient. The author needs to discuss the recently published analytical methods. The following papers may be cited and discussed in literature survey:

• Exploiting the potential of fabric phase sorptive extraction for forensic food safety: analysis of food samples in cases of drug facilitated crimes. Food chemistry, 432, 137191.

• Cellulose Paper Sorptive Extraction (CPSE) Combined with Gas Chromatography–Mass Spectrometry (GC–MS) for Facile Determination of Lorazepam Residues in Food Samples Involved in Drug Facilitated Crimes. Separations, 10(5), 281.

4. Line no 186; the author should mention the temperature at which the urine and blood samples were stored.

5. The author mustprovide the instrumental specifications of UHPLC-MS/MS and GC-MS used in determining drugs in blood and urine samples within the main manuscript.

6. What is “IQR”? The author must furnish information regarding this term for better understanding of readers.

7. The author must include the chromatograms in the main manuscript for the better understanding of the readers.

Reviewer #4: The manuscript entitled “Drugs in blood and urine samples from suspected spiked drink victims: a prospective observational study from Oslo, Norway” describes an important subject.

However, the manuscript should be corrected to be written in more standard English. There are many typos and unclear definitions (unreported drugs, spiked drinks, voluntary taken etc.), and the manuscript is generally unfocused on the aim and clear conclusions are lacking. It is unclear whether the purpose is to identify which substances are used in these cases or whether the purpose is to make a comparison of identified substances in blood and urine. The authors are encouraged to include more relevant drug classes such as sedative antihistamines, antidepressants etc. to increase the value of the study.

Words/terms which advantageously could be rephrased/changed in the manuscript:

• The authors use the term “patients” throughout the manuscript. I see the point using the term “patients” when the individuals are hospitalized or involved with the emergency medicine services. However, in some sentences it would be more correct to use individuals or victims of drink spiking. As being the case in the objective “patients regularly contact”

• Rephrase the objectives where the sentence say “patients with suspected spiked drinks”. Maybe write suspected victims of drink spiking instead.

• The authors should be consistent with the term drink spiking/spiked drink/drink-spiking throughout the manuscript. They could consider to use drink spiking and victims of dink spiking throughout the manuscript instead of spiked drink victims. Also consider to correct the title.

• Consider to rephrase “unreported drugs” as this in many sentences is confusing. Consider to use “additional drugs which was not self-reported”

• Consider to change the key-words as some are also included in the title. Use only drink spiking or spiked drinks (Depending on which definition is used throughout the manuscript). Consider to include LC-MS/MS in keywords. The keywords deviates from manuscript to the front template.

Selected comments

Abstract:

• In objectives please, clarify the aim of the study.

• The conclusion in the abstract should be changed as this does not really include all important conclusion from the paper. Could the additional drugs measured, which have not been self-reported, potentially contribute to the drink spiking cases?

Introduction:

• The introduction should be shortened and focusing more on the aim of the study.

Materials and methods:

• It is unclear whether the substances are only screened or whether they are also verified.

• Include a more detailed description on the analytical methods used.

Result:

• Line 234 rephrase: we obtained biological samples in blood.

• Table 2: Misleading numbers in the table: is total number of cases in the reported and unreported 100 in each? while the total number is also 100.

• Table 3: Why mention the symptoms and clinical features if they are not used in the classification of "drink spiking".

Discussion:

• The authors write on page 20 line 413 that antidepressants and antihistamines and other drugs not uncommonly found in DFSA cases have not been included in the study. According to SOFT and the drug-facilitated crimes (DFC) committee both antihistamines and antidepressants are among the common DFC drugs in urine samples. Maybe there would be more than 15% of the cases with additional drugs than the self-reported if these compounds have been included. Please elaborate more on that in the manuscript.

• What can this study be used for? What do the reader achieve by reading the paper?

• The study indicates that ethanol could potential be used in drink spiking, and this conclusion is based upon no additional drug findings in the samples and more alcohol in the samples with no additional drug findings. However, this cannot be proven with methods and therefore not give any answers to the victims. What about the samples with other drug findings, could they potentially contribute to the drink spiking cases? More about the actual findings are missing. And could drugs that are commonly taken (antihistamines and antidepressants?) not be used in drink spiking cases? Please elaborate on this also.

• Part conclusions contain too many non-scientific claims such as "The patient may have smokey what they thought was nicotine" line 351 and "Ethanol might have been used to spike alcoholic drinks" line 425 and section 358–371.

• Elaborate more on the ethanol results according to sampling time interval. Many of the samples were collected late, which could be the reason for lack of ethanol detection.

Conclusion:

• What about the actual drug findings, these are not really elaborated?

• Elaborate on the aim.

6. PLOS authors have the option to publish the peer review history of their article (what does this mean?). If published, this will include your full peer review and any attached files.

Reviewer #1: No

Reviewer #2: No

Reviewer #3: No

Reviewer #4: No

---

## [Author Response · Author response to Decision Letter 0]

7 Mar 2024

Dear editors and reviewers

Thank you for your valuable comments and this opportunity to revise and improve our manuscript.

REVIEWER 1

It was my pleasure to review the article titled Drugs in blood and urine samples from suspected spiked drink victims: a prospective observational study from Oslo, Norway, a solid and well researched article centered around drug-facilitated sexual assault (or less commonly other crimes targeting an uncconscious ibìncapacitated victim). The study encompasses a 100-patient sample and focuses on detection of unreported drugs.

The article has noteworthy strengths: it is thorough in terms of pursuing its stated objective; it is relevant and overall a worthy contribution to toxicology research; it has an element of novelty in its design, and relies on sound methodology as far as I was able to determine.

It is my belief that the article, particularly in its discussion, could benefit from a slightly higher degree of contextualization and broader scope when it comes to screening and detection techniques. It would certainly make the article more comprehensive and well-rounded to draw comparisons and outline distinctive features of various techniques and their analytical value in terms of tackling the spread of illegal/misused substances, both for recreational use and as a tool to victimize others. Broader remarks on the legal and law enforcement potential of detection efforts, at least within the European context, could also be advisable. The following sources should be drawn upon and cited:

 Brunetti P, Lo Faro AF, Di Trana A, Montana A, Basile G, Carlier J, Busardò FP. β'-Phenylfentanyl Metabolism in Primary Human Hepatocyte Incubations: Identification of Potential Biomarkers of Exposure in Clinical and Forensic Toxicology. J Anal Toxicol. 2023 Jan 24;46(9):e207-e217. doi: 10.1093/jat/bkac065.

 Di Trana A, Brunetti P, Giorgetti R, Marinelli E, Zaami S, Busardò FP, Carlier J. In silico prediction, LC-HRMS/MS analysis, and targeted/untargeted data-mining workflow for the profiling of phenylfentanyl in vitro metabolites. Talanta. 2021 Dec 1;235:122740. doi: 10.1016/j.talanta.2021.122740.

 Lo Faro AF, Venanzi B, Pilli G, Ripani U, Basile G, Pichini S, Busardò FP. Ultra-high-performance liquid chromatography-tandem mass spectrometry assay for quantifying THC, CBD and their metabolites in hair. Application to patients treated with medical cannabis. J Pharm Biomed Anal. 2022 Aug 5;217:114841. doi: 10.1016/j.jpba.2022.114841.

 Busardò FP, Zaami S, Baglio G, Indorato F, Montana A, Giarratana N, Kyriakou C, Marinelli E, Romano G. Assessment of the stability of exogenous gamma hydroxybutyric acid (GHB) in stored blood and urine specimens. Eur Rev Med Pharmacol Sci. 2015 Nov;19(21):4187-94.

We have elaborated on these points in the Discussion, drawing upon some of the suggested sources (lines 398-403 and 442-446, cf. also response to Reviewer 2).

The tables and figures are qite well crafted and effective at conveying key points and relevant findings.

Although the article is well-written overall, I recommend further proofreading by a native speaker of English.

We have proofread the manuscript again for English language improvement and made appropriate changes throughout.

REVIEWER 2

The article is competently assembled and straightforward enough overall. Its aim is clearly delineated and the methodology appears to be sound. tables and figures contribute to the conveyance of key elements providing substance and clarity to the article' s fundamental reasoning and conclusions.

The Discussion I believe should be further developed to better highlight the article's relevance in toxicology research. More meaningful sources ought to be used in order to better elaborate on detection techniques and their value in tackling substance abuse and in upholding public health. The following should be looked at and cited in that regard: DOI: 10.23750/abm.v92i6.12696; DOI 10.1016/j.jpba.2020.113335. 

We have elaborated on these points in the Discussion, drawing upon some of the suggested sources (lines 398-403, cf. also response to Reviewer 1, and lines 411-412), and substantiated what was previously the last paragraph in the Strengths and limitations section (now moved to the Possible drink spiking cases section) with more and appropriate sources (lines 326-332). 

Furthermore, I suggest the authors brefly address legislative aspects as well against the backdrop of the major threat posed by novel psychoactive substances DOI: 10.1002/hup.2727; DOI: 10.1007/164_2018_160; DOI: 10.26355/eurrev_201911_19529). Such additions will provide an extra degree of elaboration which will highlight the importance and value of the article's conclusions and findings.

We have addressed the issue of novel psychoactive substances more thoroughly in our revised Discussion (lines 398-403). We have also substantiated our discussion of novel psychoactive substances with the suggested sources (lines 398-403 and 327-332).

REVIEWER #3: 

1. In abstract (line no 47), Check the sentence consistency.

We have corrected the sentence.

2. Remove drink spiking word from the keyword.

We have kept drink spiking, but removed spiked drinks from the key words, cf. comment from reviewer 4, below.

3. Literature survey regarding analysis of drugs in case of drug facilitated crimes (DFC) is insufficient. The author needs to discuss the recently published analytical methods. The following papers may be cited and discussed in literature survey:

• Exploiting the potential of fabric phase sorptive extraction for forensic food safety: analysis of food samples in cases of drug facilitated crimes. Food chemistry, 432, 137191.

• Cellulose Paper Sorptive Extraction (CPSE) Combined with Gas Chromatography–Mass Spectrometry (GC–MS) for Facile Determination of Lorazepam Residues in Food Samples Involved in Drug Facilitated Crimes. Separations, 10(5), 281.

We have added this point to our literature survey in the Introduction (lines 113-114).

4. Line no 186; the author should mention the temperature at which the urine and blood samples were stored.

The temperature was 4 °C for both blood and urine. This information has been added to the Methods section.

5. The author must provide the instrumental specifications of UHPLC-MS/MS and GC-MS used in determining drugs in blood and urine samples within the main manuscript.

This has been added to the Methods section in the manuscript. 

6. What is “IQR”? The author must furnish information regarding this term for better understanding of readers.

IQR, meaning interquartile range, has now been explained on the first use of the abbreviation in both the abstract and the main text.

7. The author must include the chromatograms in the main manuscript for the better understanding of the readers.

We thank the reviewer for the suggestion. As method development is not the primary focus of this article and the methods are used on a routine basis, we do not believe adding chromatograms from three different methods will add substantially to the understanding of the topic, but rather might divert focus. 

REVIEWER #4: 

The manuscript entitled “Drugs in blood and urine samples from suspected spiked drink victims: a prospective observational study from Oslo, Norway” describes an important subject.

However, the manuscript should be corrected to be written in more standard English. There are many typos and unclear definitions (unreported drugs, spiked drinks, voluntary taken etc.), and the manuscript is generally unfocused on the aim and clear conclusions are lacking. 

We have proofread the manuscript again for English language improvement and made appropriate changes throughout. Furthermore, we have clarified the definitions (cf. our more specific responses below) and made an effort to focus the manuscript on the aim and improve the conclusion, along the lines suggested below.

It is unclear whether the purpose is to identify which substances are used in these cases or whether the purpose is to make a comparison of identified substances in blood and urine. 

The purpose was to identify the substances used in these cases, with a special concern for the unreported drugs. We have clarified this in the Objectives section both in the abstract and the main text.

The authors are encouraged to include more relevant drug classes such as sedative antihistamines, antidepressants etc. to increase the value of the study.

We agree that including these drug classes would have increased the value of the study. Unfortunately, we did not analyze for these drugs.

Words/terms which advantageously could be rephrased/changed in the manuscript:

• The authors use the term “patients” throughout the manuscript. I see the point using the term “patients” when the individuals are hospitalized or involved with the emergency medicine services. However, in some sentences it would be more correct to use individuals or victims of drink spiking. As being the case in the objective “patients regularly contact”

We have mainly kept to using the term “patients” but have made some changes to “individuals” or “victims” as suggested, or “people”.

• Rephrase the objectives where the sentence say “patients with suspected spiked drinks”. Maybe write suspected victims of drink spiking instead.

We have rephrased this to “patients suspecting exposure to spiked drinks”.

• The authors should be consistent with the term drink spiking/spiked drink/drink-spiking throughout the manuscript. They could consider to use drink spiking and victims of dink spiking throughout the manuscript instead of spiked drink victims. Also consider to correct the title.

We have tidied up our use of these terms throughout the manuscript, including in the title, now consistently using drink spiking.

• Consider to rephrase “unreported drugs” as this in many sentences is confusing. Consider to use “additional drugs which was not self-reported”

To avoid confusion, we have now provided a definition of the term “unreported drugs” in the Objectives section of both the abstract and the main text.

• Consider to change the key-words as some are also included in the title. Use only drink spiking or spiked drinks (Depending on which definition is used throughout the manuscript). Consider to include LC-MS/MS in keywords. The keywords deviates from manuscript to the front template.

We have removed spiked drinks from the key words and aligned the key words in the manuscript and the front template.

Selected comments

Abstract:

• In objectives please, clarify the aim of the study.

We have clarified the aim of the study, both in the abstract objectives and the main text objectives.

• The conclusion in the abstract should be changed as this does not really include all important conclusion from the paper. Could the additional drugs measured, which have not been self-reported, potentially contribute to the drink spiking cases?

We have added an interpretation of the significance of the unreported drugs.

Introduction:

• The introduction should be shortened and focusing more on the aim of the study.

As suggested, we have shortened the Introduction, focusing more on the aim of the study.

Materials and methods:

• It is unclear whether the substances are only screened or whether they are also verified.

Whole blood samples were analyzed by specific methods with calibrators and internal standards in line with the common criteria for quantitative methods, but were only run once. Urine samples were screened with immunoassay, while GHB in urine was run quantitatively by GC-MS. We have rewritten the manuscript to make this clear to the reader. 

• Include a more detailed description on the analytical methods used.

This has been included in the manuscript.

Result:

• Line 234 rephrase: we obtained biological samples in blood.

We have rephrased as suggested.

• Table 2: Misleading numbers in the table: is total number of cases in the reported and unreported 100 in each? while the total number is also 100.

We see that stating percentages of the total of all samples for both reported and unreported drugs did not work very well. We have changed our reporting in Table 2 as suggested.

• Table 3: Why mention the symptoms and clinical features if they are not used in the classification of "drink spiking".

We think describing the symptoms and clinical features substantiates the reasons the patients gave for suspecting drink spiking and gives an impression of the clinical picture in the presentations. Most of the symptoms and clinical features could also be due to the drugs and alcohol the patients stated they had taken. Hence, we would not expect much help from the symptoms and clinical features in discerning between having actually been exposed to drink spiking or not, cf. Table 1. 

Discussion:

• The authors write on page 20 line 413 that antidepressants and antihistamines and other drugs not uncommonly found in DFSA cases have not been included in the study. According to SOFT and the drug-facilitated crimes (DFC) committee both antihistamines and antidepressants are among the common DFC drugs in urine samples. Maybe there would be more than 15% of the cases with additional drugs than the self-reported if these compounds have been included. Please elaborate more on that in the manuscript.

We have elaborated on this in the second paragraph in Discussion (lines 323-332). This paragraph was developed from the last paragraph in the Limitations section in the previous version of the manuscript, now moved to a more prominent place to underscore this important point.

• What can this study be used for? What do the reader achieve by reading the paper?

We hope to contribute to clinicians’ understanding of the complex question of suspected exposure to drink spiking. It is of serious concern to individuals thinking themselves possibly exposed. Laboratory testing can help to some extent, though limited analytical libraries and timing issues are a problem. Furthermore, drinks may be spiked with ethanol, or with a drug also taken voluntarily by the patient. Then, there is the possibility that the patient underestimates the effects of ethanol and/or drugs taken intentionally. All these factors should be considered when addressing the concern of the patient suspecting exposure to drink spiking. We have elaborated on these implications in a paragraph added to the Conclusion.

• The study indicates that ethanol could potential be used in drink spiking, and this conclusion is based upon no additional drug findings in the samples and more alcohol in the samples with no additional drug findings. However, this cannot be proven with methods and therefore not give any answers to the victims. What about the samples with other drug findings, could they potentially contribute to the drink spiking cases? More about the actual findings are missing. And could drugs that are commonly taken (antihistamines and antidepressants?) not be used in drink spiking cases? Please elaborate on this also.

We agree that we cannot prove that ethanol was used for drink spiking in our study. We have tried to clarify this by moving the limitations paragraph on the problem of drugs we did not test for to the beginning of the Discussion (lines 323-332), by rephrasing parts of the ethanol paragraph (lines 375-391), and by elaborating on the unreported drugs found in the Conclusion.

• Part conclusions contain too many non-scientific claims such as "The patient may have smokey what they thought was nicotine" line 351 and "Ethanol might have been used to spike alcoholic drinks" line 425 and section 358–371.

We have removed the sentence about smoking, and we have toned down our claims on drink spiking with ethanol both in the ethanol paragraph in Discussion and in the Conclusion.

• Elaborate more on the ethanol results according to sampling time interval. Many of the samples were collected late, which could be the reason for lack of ethanol detection.

We have commented on this in the revised ethanol paragraph (lines 375-391).

Conclusion:

• What about the actual drug findings, these are not really elaborat

---

## [Decision Letter · Decision Letter 1]

12 Jun 2024

Drugs in blood and urine samples from victims of suspected exposure to drink spiking: a prospective observational study from Oslo, Norway

PONE-D-23-26541R1

Dear Dr. Vallersnes,

We’re pleased to inform you that your manuscript has been judged scientifically suitable for publication and will be formally accepted for publication once it meets all outstanding technical requirements.

Kind regards,

Heather M Barkholtz, Ph.D.

Academic Editor

PLOS ONE

Additional Editor Comments (optional):

Reviewers' comments:

Reviewer's Responses to Questions

**Comments to the Author**

1. If the authors have adequately addressed your comments raised in a previous round of review and you feel that this manuscript is now acceptable for publication, you may indicate that here to bypass the “Comments to the Author” section, enter your conflict of interest statement in the “Confidential to Editor” section, and submit your "Accept" recommendation.

Reviewer #1: All comments have been addressed

Reviewer #3: All comments have been addressed

2. Is the manuscript technically sound, and do the data support the conclusions?

Reviewer #1: Yes

Reviewer #3: Yes

3. Has the statistical analysis been performed appropriately and rigorously? 

Reviewer #1: Yes

Reviewer #3: Yes

4. Have the authors made all data underlying the findings in their manuscript fully available?

Reviewer #1: Yes

Reviewer #3: Yes

5. Is the manuscript presented in an intelligible fashion and written in standard English?

Reviewer #1: Yes

Reviewer #3: Yes

6. Review Comments to the Author

Reviewer #1: I have read your article and the responses to the inquiries from other reviewers with particular attention and interest. These inquiries are quite pertinent and have undoubtedly enriched the scientific content of your work. Personally, I find the impact of this study to be interesting and commend you for the scholarly presentation. In conclusion, the study certainly has interesting points as it addresses a topic of great interest. The objective is clear and adhered to. The main question addressed by the research is clear and entirely agreeable. I believe the information provided is sufficient and represents useful elements to encourage the development of new scientific work.

Reviewer #3: Manuscript Title: Drugs in blood and urine samples from victims of suspected exposure to drink spiking: a prospective observational study from Oslo, Norway

The authors have addressed my comments and now the manuscript can be accepted for publication.

7. PLOS authors have the option to publish the peer review history of their article (what does this mean?). If published, this will include your full peer review and any attached files.

Reviewer #1: **Yes: **Giuseppe Basile

Reviewer #3: No

---

## [Editor Report · Acceptance letter]

17 Jun 2024

PONE-D-23-26541R1 

PLOS ONE

Dear Dr. Vallersnes, 

I'm pleased to inform you that your manuscript has been deemed suitable for publication in PLOS ONE. Congratulations! Your manuscript is now being handed over to our production team.

Kind regards, 

on behalf of

Dr. Heather M Barkholtz 

Academic Editor

PLOS ONE